# Unbiased yeast screens identify cellular pathways affected in Niemann–Pick disease type C

Alexandria Colaco[1], María E Fernández-Suárez[1], Dawn Shepherd[1], Lihi Gal[2], Chen Bibi[2], Silvia Chuartzman[2], Alan Diot[3], Karl Morten[3], Emily Eden[4], Forbes D Porter[5], Joanna Poulton[3], Nick Platt[1], Maya Schuldiner[2], Frances M Platt[1]

Niemann–Pick disease type C (NPC) is a rare lysosomal storage disease caused by mutations in either the *NPC1* or *NPC2* genes. Mutations in the *NPC1* gene lead to the majority of clinical cases (95%); however, the function of NPC1 remains unknown. To gain further insights into the biology of NPC1, we took advantage of the homology between the human NPC1 protein and its yeast orthologue, Niemann–Pick C–related protein 1 (Ncr1). We recreated the *NCR1* mutant in yeast and performed screens to identify compensatory or redundant pathways that may be involved in NPC pathology, as well as proteins that were mislocalized in *NCR1*-deficient yeast. We also identified binding partners of the yeast Ncr1 orthologue. These screens identified several processes and pathways that may contribute to NPC pathogenesis. These included alterations in mitochondrial function, cytoskeleton organization, metal ion homeostasis, lipid trafficking, calcium signalling, and nutrient sensing. The mitochondrial and cytoskeletal abnormalities were validated in patient cells carrying mutations in *NPC1*, confirming their dysfunction in NPC disease.

## Introduction

Niemann–Pick disease type C (NPC) is an autosomal recessive lysosomal storage disorder characterized by progressive neurodegeneration. NPC is caused by mutations in either the *NPC1* or *NPC2* genes, resulting in identical clinical phenotypes irrespective of which gene is affected (1). Mutations in *NPC1* account for the majority of observed clinical cases (95%); however, the exact function of this protein remains incompletely understood. There are currently two main theories about NPC1 function; one is that NPC1 is a cholesterol transport protein that moves low-density lipoprotein-derived cholesterol out of the lysosome (2), whereas the second is that NPC1 is a cholesterol-regulated protein that is directly or indirectly involved in the transport of other lipid cargos within or across the lysosomal membrane (3). Structurally, NPC1 is a 13 transmembrane domain protein that contains a sterol-sensing domain and has structural similarities with resistance-nodulation-division permeases (multi-substrate effluxors) (4, 5). The highly conserved structure of the NPC1 protein makes it a good target for studies in simpler model eukaryotes that may provide novel insights into its conserved functions.

In the yeast *Saccharomyces cerevisiae* (here on referred to as yeast), the NPC1 orthologue is the Niemann–Pick type C–related protein (Ncr1), which localizes to the vacuole, the yeast equivalent of the mammalian lysosome (6). Studies have demonstrated that the human NPC1 and yeast Ncr1 protein are functionally equivalent, as the cellular phenotypes of patient-derived fibroblasts can be rescued through the overexpression of tagged yeast Ncr1 protein that directs it to the lysosomal membrane (6). It had previously been reported that there is no significant change in sterol or phospholipid levels in *NCR1* mutants (Δ*ncr1*), but rather a sphingolipid trafficking defect (6) where long-chain sphingoid bases (7) accumulate in Δ*ncr1* yeast. Further studies demonstrated that while sterols may not accumulate in the vacuole in Δ*ncr1* yeast (6), under starvation conditions, the processing of lipid droplets and transport of sterols to the vacuolar membrane is impaired (8). These data, implicating defects in sphingolipid and sterol trafficking, are in line with the recent structural data identifying an internal hydrophobic tunnel environment in Ncr1 that would accommodate a variety of lipids, in a capture-and-shuttle mechanism (8). This yeast Ncr1 tunnel model also further supports previous work indicating that mammalian NPC1 interacts with other sterol-shuttling proteins, including Gram1b on the ER membrane and ORP5 on the plasma membrane, and that contact sites may be necessary for lipid export from the lysosome (9, 10). Therefore, while these new models shed light on how lipids might physically move from the vacuole, the mechanisms and proteins involved in both the lipid trafficking defect and accumulation in NPC remain unknown.

[1]Department of Pharmacology, University of Oxford, Oxford, UK    [2]Department of Molecular Genetics, Weizmann Institute of Science, Rehovot, Israel    [3]Nuffield Department of Obstetrics and Gynecology, University of Oxford, Oxford, UK    [4]Institute of Ophthalmology–Cell Biology, University College London, London, UK    [5]Eunice Kennedy Shriver National Institute of Child Health and Human Development, National Institute of Health, Bethesda, MD, USA

Correspondence: frances.platt@pharm.ox.ac.uk

In this study, we exploited the power of yeast genetics and performed three independent systematic screens. Our objectives were to identify proteins that are affected by loss of Ncr1 and maybe contribute to the pathology. This could be either through a physical interaction with Ncr1, by being indirectly affected at the level of intracellular location, or by becoming essential for cellular physiology in the absence of Ncr1. Some of the genes identified in our screens are associated with cellular phenotypes reported previously in NPC disease. These include calcium dysregulation, mitochondrial dysfunction, metal ion homeostasis defects, and lipid trafficking abnormalities. However, we also identified genes involved with the cytoskeleton and nutrient sensing, biological processes not previously linked to this disorder. We found that cytoskeletal defects predicted by the yeast data occur in patient-derived cells, demonstrating the usefulness of yeast studies to further our understanding of NPC disease.

## Results

### Identification of Ncr1 interaction partners on the vacuole membrane

To shed light on the pathology of NPC using yeast as a model organism, we performed three independent, unbiased screens (Tables S1–S3). The first screen focused on uncovering additional interacting proteins for Ncr1.

NPC1 is thought to transiently interact with NPC2 to exchange cholesterol via the N-terminal cholesterol-binding loop of NPC1 in the lysosomal lumen (2). However, other interacting proteins (transient and more stable interactors) remain uncharacterized. We, therefore, performed a protein complementation assay to identify proteins that physically interact with Ncr1 on the vacuole membrane. We screened by the split-dihydrofolate reductase (DHFR) assay (11). Specifically, Ncr1 was tagged with one half of the DHFR enzyme and mated to strains carrying fusion proteins to the other half of the enzyme. Only if proteins physically interact will this enable the complementation of the full DHFR enzyme and resistance to methotrexate, which inhibits the endogenous, essential, DHFR but not the synthetically encoded one. Because the DHFR fragments in this library are fused at the carboxyl terminus (C′), we assayed only the 48 proteins that are known to localize correctly to the vacuole membrane when tagged at their C′ (12) (Table S1). Because only vacuolar membrane proteins were used in the screen, known interactors such as Npc2, which resides in the vacuole lumen, were not observed. Of the 48 vacuolar membrane proteins assayed, only three interacted significantly: Pmc1—a calcium ATPase, Apc11—an anaphase-promoting complex member involved in cell cycle regulation, and Fth1—an iron transport protein. The strength of their interactions with Ncr1 was measured as a function of colony size divided by their abundance and was calculated to be 0.4, 0.2, and 0.15 for Pmc1, Apc11, and Fth1 respectively, posing Pmc1 as the most robust vacuolar membrane interactor of Ncr1 (Fig 1A).

To validate the interaction with Pmc1, we performed a reciprocal assay, using Pmc1 as the bait. This screen verified the interaction with Ncr1, thereby confirming the physical association between the two

proteins. Strong interactions were again identified with the anaphase-promoting complex protein Apc11 and the iron transporter Fth1. In addition to these, we identified the ABC transporter Ycf1 as a strong interacting partner with Pmc1 (Fig 1B). Based on these data of interaction both between Pmc1 and Ncr1, as well as other shared protein interaction partners Apc11 and Fth1, it suggests that these proteins may be residing as a complex on the vacuolar membrane (Fig 1C). The interaction between Ncr1 and Pmc1 had also been previously suggested from a genome-wide protein–protein interaction screen (11); however, this interaction had not been validated.

Because our screens aimed to shed light on the mammalian NPC1 protein, we investigated the interaction of NPC1 with the human homologue of Pmc1, PMCA2/ATP2B2 in a mammalian system. We performed a co-immunoprecipitation from isolated rat cerebellum and demonstrated that PMCA2/ATP2B2 is pulled down together with NPC1, validating the yeast screen findings (Fig 1D). Furthermore, we examined Pmca2 (Atp2b2) transcript levels in the $NPC^{m1n}$ mouse model, which is null for the Npc1 protein because of an insertion resulting in deletion of 11 of the 13 transmembrane domains resulting in a premature truncation of the Npc1 protein (13). A significant reduction in mRNA levels of Pmca2/Atp2b2 in the $Npc1^{-/-}$ mouse cerebellum as compared with the WT ($P = 0.0004$, Fig 1E) was observed. This suggests that the loss of functional NPC1 protein could affect the expression of proteins, such as PMCA2, that it interacts with.

### Pathways essential to sustain viability in Δncr1 yeast corroborates a role for mitochondria in NPC

Changes in protein localization may result from primary or secondary loss of cellular function and abnormal physiology in the absence of Ncr1 or they may reflect a compensatory mechanism(s). For example, cholesterol accumulation and impaired transferrin receptor trafficking in NPC1-deficient CHO cells can be corrected by overexpression of acid sphingomyelinase, whose reduced activity is a secondary defect in these cells (14). To differentiate between these options, we hypothesized that pathways that compensate for loss of Ncr1 would be essential for survival of the Δncr1 strain. We, therefore, performed a synthetic sick/lethal screen that compared a genome-wide yeast knockout library crossed onto the Δncr1 background, with the same library crossed to a wild-type background. This systematic synthetic sick/lethality screen identified more than 50 proteins whose loss exacerbated the phenotype of Δncr1, resulting in slower growth (Table S2).

Genes that were identified in this screen included those involved in copper transport (*MAC1*), sphingolipid and fatty acid biosynthesis (*FEN1* and *HTD2*), mitochondrial function (*FZO1* and *ILM1*), and protein sorting (*MVB12*) (hits found summarized in Table 1). To confirm these results, a serial dilution was performed on a Δncr1 background with either *MAC1* or *FZO1* under a repressible promoter (*GAL1pr*) (Fig 2A). Indeed, growing the cells on glucose to repress transcription of *MAC1* demonstrated a more severe growth phenotype on the background of double mutants relative to the loss of Ncr1 alone. However, *FZO1* expression is essential for normal yeast growth, so in fact, repressing *FZO1* expression on the Δncr1 background enhanced its growth, suggesting that the proteins operate in reciprocal pathways (Fig 2A).

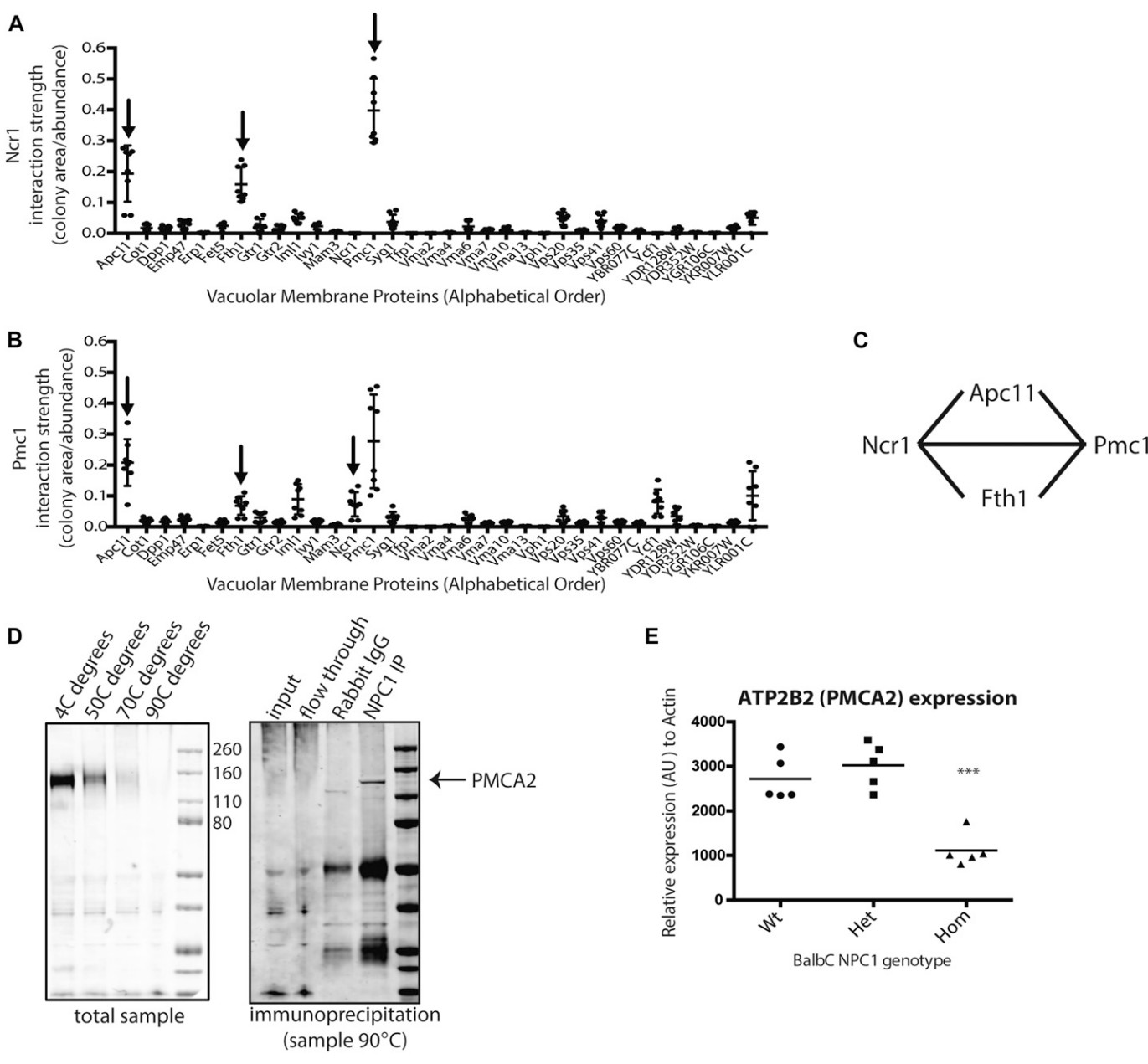

**Figure 1.  Identification of Ncr1 interaction partners on the vacuole membrane.**
**(A)** Ncr1 was tagged with one half of the DHFR enzyme and mated to strains carrying fusion proteins to the other half of the enzyme. Interaction strength was a function of colony size of the diploids on methotrexate divided by their abundance. Ncr1 had high interaction strength with Pmc1 and Fth1 relative to the panel of vacuolar proteins. **(B)** Pmc1 was tagged with one half of the DHFR enzyme and mated to strains carrying fusion proteins to the other half of the enzyme and had high interaction strengths with Fth11, Apc11, and Ncr1 relative to the panel of vacuolar proteins. Means ± SD, N = 8. **(C)** Graphic of predicted complex of Ncr1, Pmc1, Apc11, and Fth1. **(D)** Co-immunoprecipitation of NPC1 and PMCA2 (ATP2B2) from rat cerebellum. **(E)** qRT-PCR for mRNA expression of PMCA2 (ATP2B2) in wild-type, heterozygous, and homozygous cerebellums of 8-wk Npc1[nih] balb/c mice. N = 5, ***$P$ = 0.0004 as compared with WT, calculated by one-way ANOVA.

Genes associated with mitochondrial function were identified in both the mis-localization screen and the synthetic sick/lethal screen. Interestingly, although an abundance of mitochondrial genes were indicated, we did not observe any gross defects in mitochondrial morphology or function in the Δ*ncr1* yeast by growing colonies on glycerol, a non-fermentable carbon source (Fig S1A and B), suggesting any phenotypes may be quite subtle and compensated for by multiple mitochondrial pathways. In light of this, and

the expanding body of data demonstrating impaired mitochondrial homeostasis in NPC disease (15, 16, 17, 18), we followed up our findings made in yeast by investigating relevant mitochondrial phenotypes in NPC1 patient cells.

One of the strongest hits identified in the synthetic sick/lethal yeast screen was *ILM1*, a gene essential for mitochondrial DNA (mtDNA) maintenance (19). We, therefore, quantified mtDNA in NPC1 patient cells and observed a significant reduction (>20%) as

**Table 1.** Specific proteins and genes extracted from yeast Ncr1 screens according to function or organelle association.

| Function | Screen for interaction partners | Mislocalisation screen | Synthetic sick/lethal screen |
|---|---|---|---|
| Mitochondrial function | | Rcf1, Mcy1, Mrpl23, Aep1, Mss116, Nam2, Rsf1, and Coq3 | *SOM1*, *FMP37*, *MCT1*, *MRPL17*, *COA2*, *ATP20*, *ILM1*, *RSM22*, *STF2*, *HTD2*, *FZO1*, *ADK1*, *OCT1*, and *MSF1* |
| Cytoskeleton | | She4 and Prk1 | *ADA2* |
| Calcium homeostasis | Pmc1 | Pmr1 | |
| Metal ion homeostasis | Fth1 | Ctr1 | *MAC1* |
| Lipid trafficking | | Pry1 | |
| Nutrient sensing and nutrient uptake | | Tco89 | *AGP1* |
| Protein sorting | | Vps41, Did4, Pmr1, and Yip5 | *MVB12* and *APS1* |
| Peroxisome | | Pex17 | *AAT2* |
| Lipid homeostasis | | Erg27 and Sel1 | *CHO2* and, *FEN1* |
| Cell cycle | Apc11 | | *WHI4* |

compared with healthy controls ($P < 0.001$; Fig 2B). In addition, we found hyperpolarization of the mitochondrial membrane potential in NPC1 patient fibroblasts ($P < 0.0001$; Fig 2B) and an increase in perinuclear mitochondria in the NPC1 fibroblasts ($P < 0.01$; Fig 2B). Moreover, when we measured the rate of oxygen consumption in WT and NPC1-deficient CHO cells, we found that NPC1 null cells grown in 1 $\mu$M reduced glucose medium displayed significantly reduced oxygen consumption as compared with controls ($P < 0.0001$; Fig S1C).

Mitochondrial length is determined through a balance of fission and fusion events (20). Interestingly, a gene identified in the synthetic sick/lethal screen, the GTPase *FZO1* (mitofusin), is required for mitochondrial fusion (21, 22). Using high content-screening (23) together with electron microscopy (Fig 2C), we determined mean mitochondrial length in NPC1 patient cells and confirmed that there was a significant increase in length in comparison with controls (GM3123, 2.59 $\mu$m ± 0.077; $NPC1^{+/+}$, 1.74 $\mu$m ± 0.043 **$P < 0.01$, n = 100; Fig 2D). Because the fusogen *FZO1* is essential for growth in the absence of Ncr1, it suggests that hyper fusion is a protective mechanism that cells rely on in the absence of Ncr1/NPC. Furthermore, deletion of the fission gene (FIS1) in Δncr1 *GAL1-FZO1* double-mutant cells partially rescued the phenotype, suggesting that it is the imbalance of mitochondrial fission and fusion that is causing the lethality (Fig 2E).

## Alterations in protein localization caused by loss of Ncr1 lead to cytoskeleton abnormalities in NPC

We hypothesized that the Δncr1 strain, which does not show a significant growth phenotype, has modified or re-organized cellular networks to compensate for the loss of function of Ncr1. Using a library in which each protein is tagged with GFP (12), we used automated mating approaches (24) to integrate the Δncr1 allele into the GFP library and, using a high-content microscopy setup (25) systematically compared the localization of all yeast proteins, when expressed on the Δncr1 background compared with control colonies.

We found that more than 40 proteins were localized to a different organelle in Δncr1 as compared with control strains (Table S3). Although some may be a direct effect of Ncr1 loss, others may be the result of adaptive changes to altered cellular physiology. Proteins that were found to have an altered organelle localization included those involved in copper sensing and regulation (Ctr1), vacuole protein sorting (HOPS and ESCRT: Vps41 and Did4), myosin motors (She4p/Dim1), mitochondrial respiration (Rsf1 and Rcf1), sterol transport (Pry1), peroxisome biogenesis (Pex17), nutrient sensing (Tco89), and actin cytoskeleton organization (She4 and Prk1) (cytoskeletal proteins Prk1, She4 shown in Fig 3A, hits summarized in Table 1).

As we identified proteins associated with actin organization (Prk1 and She4) to be mislocalized in the GFP screen, we hypothesized that as mitochondria move along actin filaments in budding yeast (26), cytoskeletal dysfunction could be contributing to the mitochondrial dysfunction observed in the patient fibroblasts. In higher eukaryotes, mitochondrial positioning is regulated by microtubules (27) (rather than actin in yeast), so we examined the microtubule network in NPC-deficient mammalian cells.

One of the genes identified in the synthetic sick screen, *ADA2*, has been shown in yeast to potentiate the acetyltransferase activity of Gcn5 (28). As mammalian GCN5 plays a role in $\alpha$-tubulin acetylation (29) and mitochondria in mammalian cells move preferentially on acetylated microtubules (30), we first examined the acetylation level of the tubulin network in NPC1 patient fibroblasts compound heterozygous for the most common human mutation I1061T, and also R1186G. Patients with this mutation encode a functional NPC1 protein, but it is then targeted for degradation due to protein misfolding (31). We examined the relative amount of acetylated $\alpha$-tubulin and observed a significantly higher level of acetylated $\alpha$-tubulin staining fluorescence in the NPC1 patient cells as compared with controls (representative images Fig 3B; $P < 0.01$ Fig 3C).

Western blotting confirmed the enhanced levels of acetylated $\alpha$-tubulin, together with greater amounts of $\alpha$-tubulin acetyltransferase ($\alpha$TAT1) (Fig 3D and E). However, interestingly an increase in both the deacetylases, sirtuin 2 (SIRT2) and histone

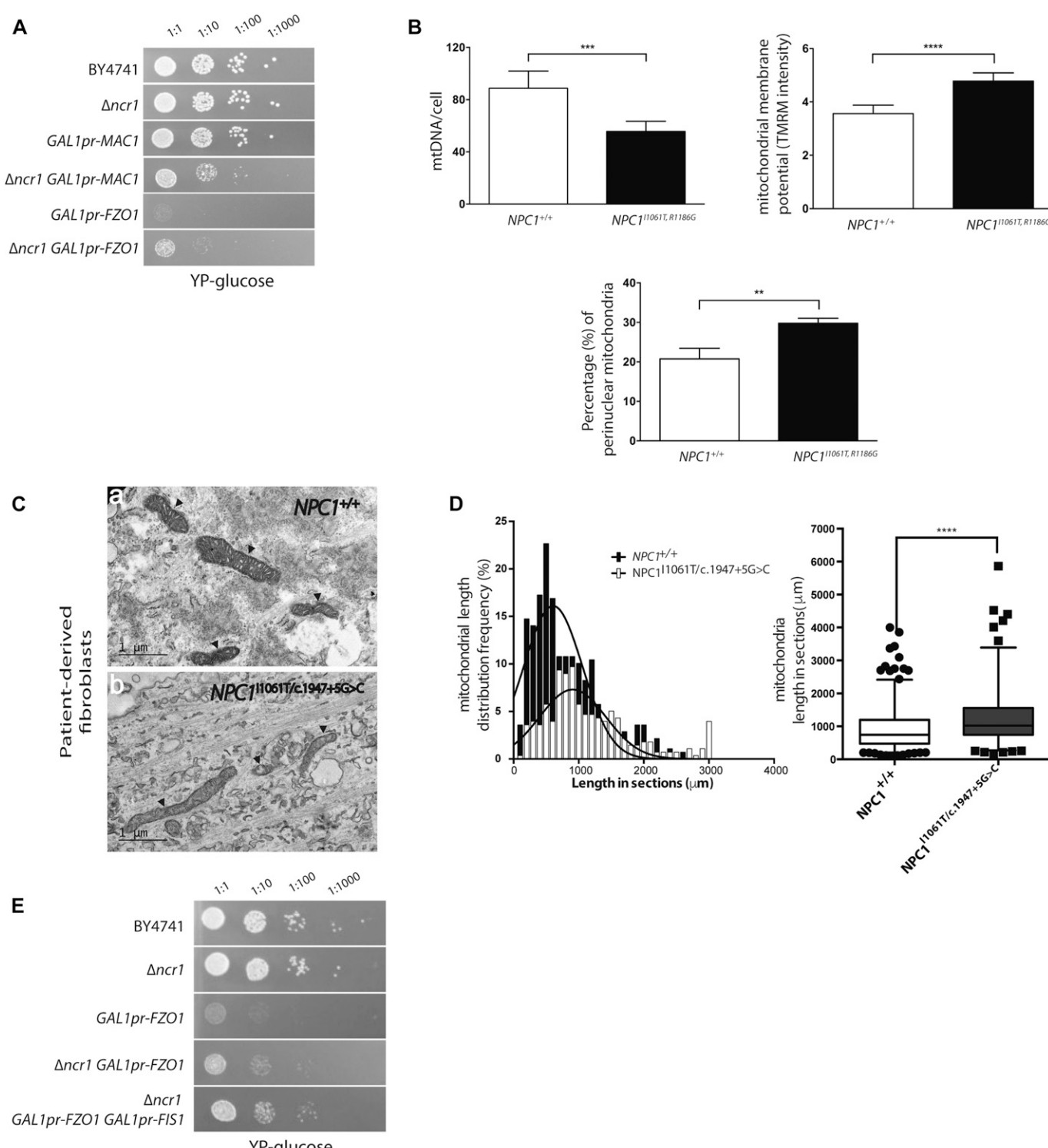

**Figure 2. Pathways essential to sustain viability in ∆Ncr1 yeast corroborates a role for mitochondria in NPC.**
**(A)** Serial dilution of yeast strains was performed on ∆ncr1 background with either *MAC1* or *FZO* on *GAL1pr* repressible promoter and grown in glucose to inhibit protein expression. **(B)** The relative mitochondrial DNA (mtDNA) content, TMRM intensity, and perinuclear distribution of mitochondria were measured in control and NPC patient cells. Mean ± SD. N = 100 **P < 0.01, ***P < 0.001, ****P < 0.0001; *t* test. **(C)** Representative EM images of control (a) and NPC (b) mitochondria. Scale bar = 1 *μ*m. **(D)** Statistical distribution of the length in sections of control *NPC1*[+/+] (open columns) and *NPC*1[−/−] patient (filled columns) mitochondria acquired from analyses of EM data. ****P < 0.0001, calculated by unpaired *t* test with Welch's correction. **(E)** Serial dilution of ∆ncr1 on *GAL1pr-FZO, FIS1* background in glucose to inhibit protein expression.

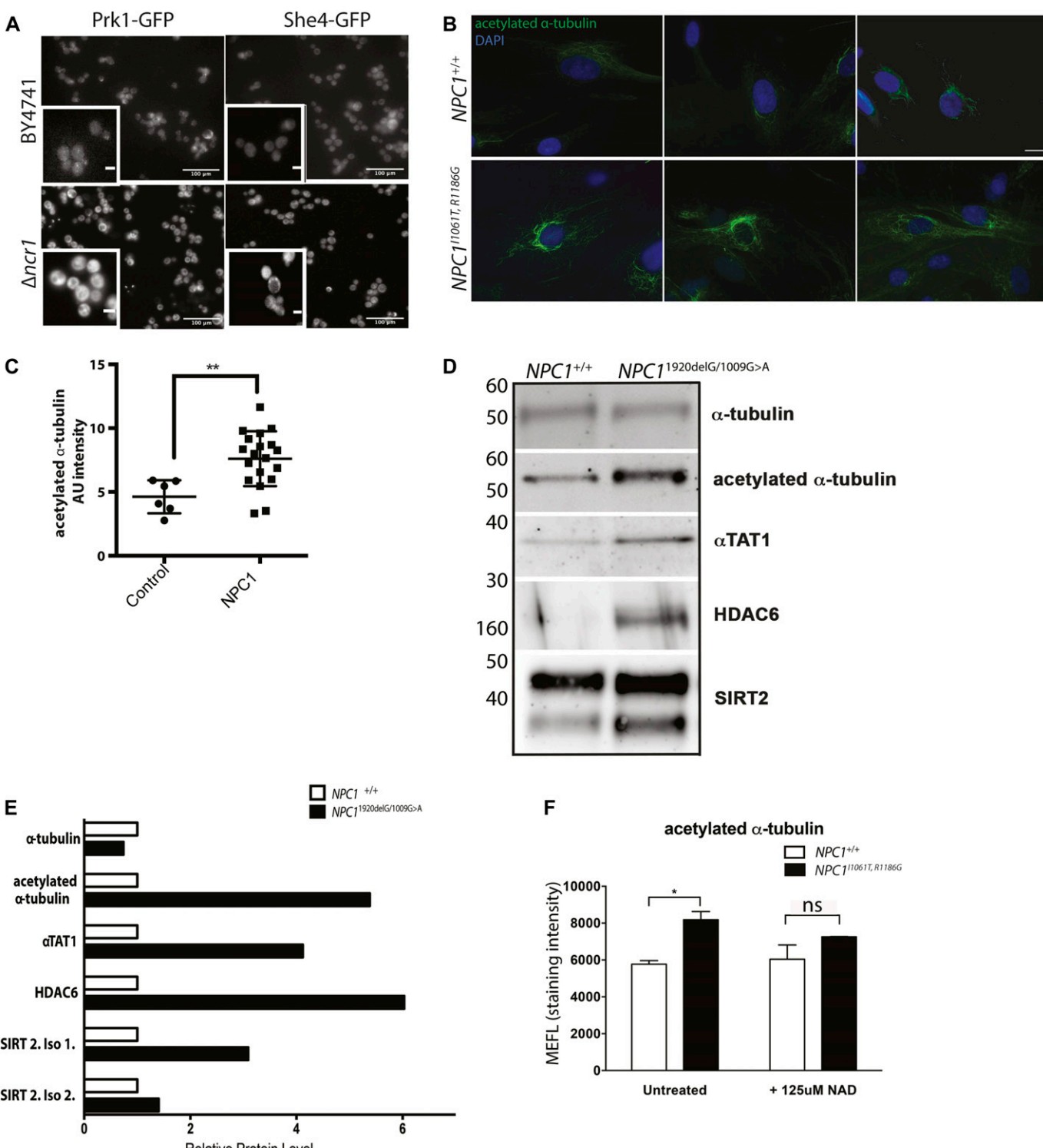

**Figure 3. Alterations in protein localization caused by loss of Ncr1 lead to cytoskeleton abnormalities in NPC.**
**(A)** Representative images from GFP localization screen including Prk1-GFP and She4-GFP in control (BY4741) relative to Δncr1 yeast. Prk1 is localized to the actin cortical patch in BY4741 yeast, but in the Δncr1 yeast Prk1 is found to be diffuse and mainly cytoplasmic. She4 is localized primarily to the cytoplasm where in the Δncr1 yeast, the localization is more punctate. **(B)** Representative images of NPC1 and control patient fibroblasts stained for acetylated α-tubulin. Scale bar: 10 μm. **(C)** Graphical representation of the fluorescent intensities of acetylated α-tubulin. Mean ± SD. N = 10–20 **P < 0.01; t test. **(D)** Protein levels of α-tubulin, acetylated α-tubulin, αTAT1, HDAC6, and SIRT2 were measured in control and NPC1 patient fibroblasts. **(D, E)** Quantitation of blots from (D) by densitometry. **(F)** Treatment with 125 μM NAD for 24 h reduced acetylated α-tubulin expression in NPC1 patient cells, fluorescence measured using a flow cytometer and analysed with FloJo software. Mean ± SD, N = 3 *P < 0.05; one-way ANOVA.

deacetylase 6 (HDAC6) (Fig 3D and E), was also observed, suggesting that the cells may be attempting to compensate for the increase in acetylated α-tubulin in an attempt to restore homeostasis. This suggests that the hyperacetylation is a consequence of NPC1 loss but is not favourable to the cell, highlighting this phenotype as a potential target for therapeutic intervention. We postulated that further enhancement of SIRT2 activity might reverse hyperacetylation of α-tubulin in NPC1 cells. We, therefore, treated patient cells with NAD, a co-factor of SIRT2, and observed normalization of the levels of acetylated α-tubulin (Fig 3F), demonstrating this hyperacetylation can be normalized.

### NPC therapies significantly reduce levels of acetylated α-tubulin in disease fibroblasts

To determine whether therapeutic interventions could also reverse he acetylated microtubule phenotype in NPC disease cells, we treated NPC1 patient fibroblasts with the European Medicines Agency-approved drug miglustat, an inhibitor of glycosphingolipid biosynthesis that provides benefit via substrate reduction (32, 33), and hydroxypropyl-β-cyclodextrin (HPβCD), a cholesterol-sequestering agent for which results from a phase I/II clinical trial in NPC1 have been reported (34). Indeed, hyperacetylation was significantly reduced after treatment with both miglustat and HPβCD, so that levels of acetylated a-tubulin were not significantly different from control cells (+50 μM miglustat P > 0.9999, +250 μM HPβCD P = 0.3003) (Fig 4A and B), supporting the hypothesis that the acetylation is a downstream effect of losing NPC1 activity. Moreover, our results demonstrate cytoskeletal defects from actin in yeast to microtubules in humans, suggesting cytoskeletal defects may be novel hallmarks of NPC1.

## Discussion

We have investigated novel aspects of the cellular features of NPC disease by taking advantage of the highly conserved yeast Ncr1 protein orthologue of NPC1. NPC1 is an evolutionarily conserved protein, with orthologues identified in both yeast and prokaryotes (13) that share homology with bacterial permeases of the resistance-nodulation-division family (35). Functionally, it has been established through the experimental demonstration that the yeast Ncr1 can rescue the cellular phenotypes of $NPC1^{-/-}$ patient fibroblasts (6) and that the mammalian protein has pump activity when expressed in

*Escherichia coli* (36). A number of studies of Ncr1 have also extended our understanding of the mammalian protein (6, 7, 37, 38) by examining and detecting phenotypes in deletion strains.

To further the insights derived from this model organism, we performed unbiased genetic screens in yeast with the confidence that these assays were likely to yield data relevant to the biology of mammalian NPC1 and elucidate which protein interactions were relevant in contributing to the cellular phenotypes. We used three separate strategies to identify proteins that might have direct or indirect interactions with Ncr1: a complementation assay to detect proteins that physically interact with Ncr1 on the vacuolar membrane, a localization screen for proteins that are mislocalized when expressed on an Ncr1-deficient background, and a synthetic sick/lethal screen to distinguish proteins that may function in compensatory or parallel pathways and whose loss leads to impaired growth or lethality in Δncr1 yeast.

We identified three proteins (Pmc1, Apc11, and Fth1) that displayed a robust interaction with Ncr1 on the vacuole membrane, more than 40 proteins that changed their localization pattern in Δncr1 background and more than 50 proteins whose presence is required for normal growth in the absence of Ncr1 (Tables S1–S3). The genes and proteins implicated from the yeast screens fell into a number of discrete categories: trafficking, nutrient sensing, calcium and metal ion regulation, mitochondrial function, and cytoskeleton–the three last ones of which we followed up on.

We have previously reported that NPC is a disorder involving altered lysosomal calcium homeostasis (39), and this is due to the accumulation of sphingosine that either inhibits lysosomal calcium uptake or promotes calcium leak. However, the identity of the protein(s) responsible for refilling the lysosome with calcium remains elusive.

The calcium ATPase Pmc1 was previously shown in a genome-wide screen for protein–protein interactions to interact with Ncr1 (11), the confirmation of this interaction on the yeast vacuole is of particular interest as its mammalian orthologues, ATP2B1-4, are members of a family of plasma membrane ATPase (PMCA) calcium transporters (40). Interestingly, it has been demonstrated that sphingomyelin accumulation impairs PMCA activity, causing loss of calcium homeostasis, oxidative stress, and neurodegeneration (41). Storage of the same lipid occurs in NPC (39), as does loss of calcium homeostasis, oxidative stress, and neurodegeneration. These findings suggest that mutations in the NPC1 protein may in turn have effects on the function of proteins that it physically interacts with. Furthermore, although the identification of the

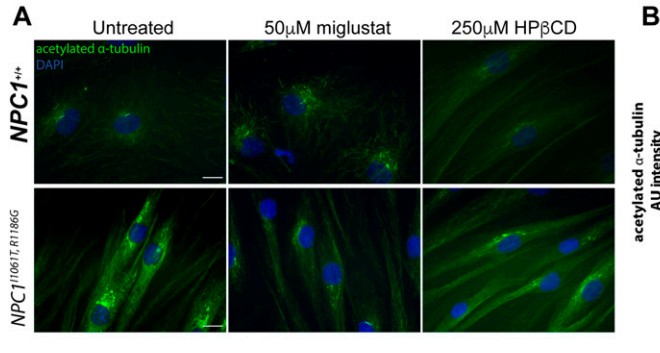
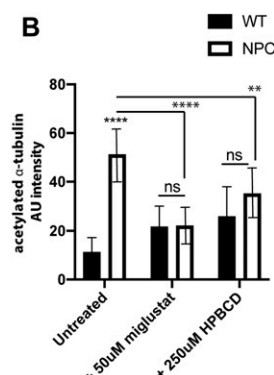

**Figure 4. NPC therapies significantly reduce levels of acetylated α-tubulin in disease fibroblasts.**
**(A)** Representative images of control and NPC1 patient fibroblasts treated with 50 μM miglustat or 250 μM HPβCD stained for acetylated α-tubulin. Scale bar: 10 μm. **(B)** Quantification of the fluorescent intensities of acetylated α-tubulin. Mean ± SD. N = 10–20 ****P < 0.0001, ns P > 0.05 two-way ANOVA.

pumps and/or transporters that may fill the lysosome with calcium remains unknown, ATP2B2 is a potential candidate and its localization should be further studied.

In addition to the identification of proteins involved in calcium homeostasis, the identification of multiple yeast proteins that are associated with mitochondria was of particular interest. These findings are in agreement with previous studies (7) that reported that loss of Ncr1 resulted in mitochondrial dysfunction as indicated by increased oxidative stress, shorter lifespan, reduced oxygen consumption, and lower cytochrome c oxidase activity. We, therefore, further investigated these mitochondrial defects in patient-derived fibroblasts and confirmed that NPC1 cells displayed lower mtDNA content, which may in fact underlie the impaired oxygen consumption observed in CHO cells. Mutations in mtDNA are known to be associated with a wide range of clinical mitochondrial diseases and a high percentage of these are located in mitochondrial tRNA genes (42). Intriguingly, the mitochondrial tRNA protein Nam2 was mislocalized in Δncr1 yeast, and mutations in the human orthologue (LARS2) are associated with conditions that involve increased cell death and multisystem failure (43).

In addition, mitochondria from NPC1 patient fibroblasts had increased length and were hyperpolarized in comparison with control fibroblasts. Although a previous report (18) suggested that mitochondria in NPC cells were smaller than those of controls, no quantitative data supporting this conclusion was provided in that study. We measured mitochondrial length using electron microscopy and observed a significant increase in length. It has been proposed that increased mitochondrial fusion can help overcome low levels of stress (44), thus this increase in length suggests that the loss of Ncr1 causes stress-induced mitochondrial fusion. This stress-induced mitochondrial fusion is a response in which mitochondria raise cellular ATP in response to various insults by elongating and hyperpolarizing (45). It has also been suggested that mitochondrial fusion is a protective measure against mtDNA loss/mutants, and enhanced fusion may be a compensation mechanism to the reduced mtDNA levels (46). Previous studies have, however, observed fragmented mitochondria in neurodegenerative diseases (47), so taken together with our data, this suggests that it is most likely a disturbance to the fission/fusion dynamic balance that results in mitochondria dysfunction.

These findings add to an expanding body of data showing loss of mitochondrial homeostasis in NPC disease (15, 16, 48). Although we do not have a complete understanding of the mechanistic link between lysosome and mitochondrial dysfunction in NPC, there is intimate communication between the two organelles that enables functional crosstalk (49, 50, 51). Data also suggest the possibility that impairment of lysosomal function may be further amplified by feedback from dysfunctional mitochondria (52). Especially as in NPC disease cells, there is an increase in lysosome–mitochondria membrane contact sites (10), which may contribute to the increased mitochondrial cholesterol (53) and mitochondrial dysfunction in NPC.

The novel identification in the screens of yeast genes associated with the cytoskeleton, such as PRK1, a serine/threonine kinase that regulates the organization of actin (54) and SHE4, a regulator of myosin motor domains (55), suggest that impaired mitochondrial function in NPC could result from defective organelle transport. We, therefore, investigated whether there are cytoskeletal phenotypes in mammalian NPC cells, specifically the microtubule network as this is key in regulating the movement of mitochondria in mammalian cells. Although in this study we focused on microtubules, other

alterations to the cytoskeleton in NPC mammalian cells have been described. For example, vimentin, a phosphoprotein component of intermediate filaments, is hypo phosphorylated in NPC, leading to its disorganization and disruption of vesicular transport (56).

The dynamic properties of microtubules are known to be affected by posttranslational modification of tubulin subunits (57), and we observed significantly increased levels of acetylated α-tubulin in patient fibroblasts. The precise role of α-tubulin acetylation in the control of microtubule dynamics has not been fully resolved (58), but increased acetylation is a hallmark of stabilized microtubules (59) and promotes flexibility (60), which increases organelle resistance to mechanical stress (61).

Although the acetyltransferase αTAT1 is responsible for acetylation (62), the major regulators of deacetylation of α-tubulin at lysine 40 are histone deacetylase 6 (HDAC6) (63) and sirtuin 2 (SIRT2) (64); loss of either leads to hyperacetylation (64, 65), which we observed. We were able to confirm increased levels of all three regulators, consistent with gene expression data (38), which may be indicative of attempts to restore α-tubulin acetylation homeostasis.

The demonstration, here and by others (38), of increased expression of HDAC6 in NPC cells is relevant because of the interest in the use of HDAC inhibitors as treatments for NPC. Several studies (66) have shown that the drug vorinostat (suberoylanilide hydroxamic acid, SAHA, Zolinza) can normalize liver lipid homeostasis in NPC models. Vorinostat does not inhibit SIRT2, so would not affect the deacetylase activity of the protein, but it does reduce HDAC6 expression (67). In light of this, the relationship between microtubule acetylation and organelle (both mitochondrial and lysosomal) dysfunction deserves further examination, as it is possible that HDAC inhibitors could aggravate this particular NPC cellular phenotype.

In summary, we have exploited genetic methods in a simple model organism to identify proteins that interact—physically or genetically—with the yeast orthologue of NPC1, and that are likely to be relevant to the functioning of the mammalian protein. What is clear from the yeast data is that Ncr1 likely acts as part of a protein complex with direct and indirect binding partners. This raises the possibility that some mutations in the NPC1 protein may not affect the primary function of this protein per se but prevent it from binding/interacting with other proteins that are required for the complex to function as a whole. This will be an interesting area of future investigation and will shed light on how NPC1 and its interacting partners function in calcium and iron regulation, organelle trafficking, mitochondrial function, cytoskeleton organization, and nutrient sensing, as well as point to new potential therapeutic avenues for the treatment of NPC.

# Materials and Methods

### Yeast strains and libraries

We performed three independent, unbiased screens detailed below. Genes that were identified are listed in Tables S1–S3. Known or proposed activities for the proteins they encode are listed according to the *Saccharomyces* Genome Database (https://www.yeastgenome.org).

All yeast strains in this study are based on the BY4741 laboratory strain (68). Gene deletion was performed using the pFA6 plasmid series and verified using PCR for the loss of the gene copy (69). GFP-tagged

strains were picked from the GFP library (12) and deletion strains were picked from the yeast deletion library (70). Known or proposed activities for the proteins they encode are listed according to the *Saccharomyces* Genome Database (https://www.yeastgenome.org).

## Robotic library manipulations

All genetic manipulations on entire libraries were performed using Synthetic Genetic Array (SGA) techniques (24, 71). To manipulate libraries in 1536-colony high-density format, a RoToR bench top colony arrayer (Singer Instruments) was used.

## Yeast library screening

The Δ*ncr1* strain was constructed and mated against the yeast deletion library (70) using SGA techniques that allow efficient introduction of a trait (mutation or marker) into systematic yeast libraries. SGA was performed as previously described (24, 71). Colony size was then quantified using the Balony free software for the analysis of images of plates containing arrays of yeast (the software package is maintained by Barry Young at the University of British Columbia, Vancouver, Canada). For the GFP library screen, the colonies were moved to liquid medium, and for high-throughput microscopic screening.

## High-throughput microscopy

Microscopic screening was performed using an automated microscopy system (ScanR system; Olympus) as previously described (24). Briefly, images were acquired for GFP (excitation 490/20 nm; emission 535/50 nm) and bright-field channels. After acquisition, the images were manually reviewed in MATLAB vs.2012a 7.17 using compare2picturesV5 script. As there were no co-localization markers, we assigned only those localizations that could be easily discriminated by eyes.

## Protein complementation assay screen using the DHFR library

Strains were taken from the DHFR protein complementation libraries (11). In this library, Haploid strain "a" is Nat resistant (+Nat), whereas haploid strain "α" is hygromycin B (+Hygro) resistant. Pmc1 haploid strain "a" and "α" were used; however, because of no Ncr1 haploid strain "a," only Ncr1-α was used. Haploid strains of either pmc1 or ncr1 were mated with the vacuolar membrane protein library on YPD-rich media plates (n = 8). Diploid selection was done twice on plates containing selection markers (+Nat, +Hygro). Diploids were then moved to metallux media (0.200 g Methotrexate + 20 ml DMSO +YPD) for 7 d to select for proteins that are physically interacting. Colony size was then quantified using the Balony free software for the analysis of images of plates containing arrays of yeast (the software package is maintained by Barry Young at the University of British Columbia, Vancouver, Canada). Interaction strength was calculated by dividing colony size by relative abundance.

## Cells

Human NPC1-mutant fibroblasts were obtained from the National Institute of Health (NPC5; severity score 14, 1061T/R1186G) and from Coriell Institute for Medical Research (GM22871 [1920delG/IVS9-1009G>A]). Control human dermal fibroblasts were acquired from Sigma-Aldrich

(Cat. no. 106-05A). The fibroblasts were maintained in DMEM with 10% FCS, 1% penicillin/streptomycin, and 1% L-glutamine. All cells were cultured at 37°C with 5% $CO_2$. Antibodies and reagents were sourced as follows: mouse anti-acetylated α-tubulin (6-11B-1; Santa Cruz Biotechnology), mouse monoclonal anti–α-tubulin (ab11304; Abcam). Quant-iT PicoGreen sdDNA (Thermo Fisher Scientific) tetramethylrhodamine methyl ester (TMRM; Invitrogen).

## Co-immunoprecipitation

Rat brain tissues were homogenized, in NP-40 cell lysis buffer (Invitrogen) supplemented with Halt Protease Inhibitor Cocktail (Thermo Fisher Scientific) using an electric homogenizer and passed through a 25-gauge needle 10 times, followed by constant agitation for 2 h at 4°C. The lysates were cleared by centrifugation at 16,000*g* for 20 min at 4°C. 500 μl of each supernatant was incubated, overnight at 4°C, with SureBeads Protein A magnetic beads (Bio-Rad) previously bound to 10 μg of NPC1 (NB400-148; Novus Bioscience) rabbit polyclonal antibody or 10 μg of IgG from rabbit serum (Sigma-Aldrich), following the manufacturer's instructions. The beads were washed and eluted by incubation, at an indicated temperature for 10 min, with lysis buffer supplemented with Protein Loading Buffer.

## Mitochondrial morphology quantification

Using the high-content IN Cell 1000 (500 cells acquired per well; GE Healthcare Life Sciences) analyzer, we quantitatively measured fluorescent mitochondria labelling in control and NPC1 fibroblasts. Raw images were processed, and parameters were obtained using a customized protocol in the IN Cell developer toolbox (GE Healthcare Life Sciences) (23).

## mtDNA analysis

DNA was purified using the DNeasy blood and tissue kit (QIAGEN) and amplified on a Corbett real-time quantitative PCR machine using Taqman universal PCR master mix. Sequences of the mitochondrial primers (100 nM) were AGGACAAGAGAAATAAGGCC and TAAGAAGAG-GAATTGAACCTCTGACTGTAA and probe (6-FAM; 200 nM) TTCACAAAG-CGCCTTCCCCCGTAAATGA. Copy number was normalized to the levels of the single copy nuclear gene amyloid β, using primers (300 nM) TTTTTGTGTGCTCTCCCAGGTCT and TGGTCACTGGTTGGTGGC and probe (200 nM; Yellow Yakima) CCCTGAACTGCAGATCACCAATGTGGTAG.

## Cytoskeleton image acquisition

All image acquisition was completed with Leica TCS SP8 scanning laser confocal microscope equipped with LAS X software. Z-stack image series were projected for maximum intensity with Fiji-ImageJ software (National Institute of Health; version 1.46), and contrast/brightness for applied Look Up Tables for each channel were applied from untreated, wild-type fibroblasts.

## Measurement of oxygen consumption

Control and NPC1-deficient CHO cells (39) were maintained in complete DMEM-F12 growth medium and before analysis transferred to the medium containing reduced levels of glucose for 24 h. Oxygen

consumption was measured using the MitoXpress-Xtra assay (Luxcel) according to the manufacturer's instructions. Readings were made at 1 min 30 s intervals for 13 h. Data were standardized against cell number determined by propidium iodide staining.

## Western blotting

Cell lysates and co-immunoprecipitated samples were separated on 4–12% SDS–PAGE gels (Thermo Fisher Scientific) and then transferred to membrane using Bio-Rad Trans-Blot Turbo system. Membranes were blocked; incubated with primary antibodies—acetylated $\alpha$-tubulin (6-11B-1; Santa Cruz Biotechnology), ATAT1 (ab58742; Abcam), SIRT2 (12650; Cell Signalling Technology), HDAC6 (H-300; Santa Cruz Biotechnology), PMCA2 (PA1-915; Thermo Fisher Scientific), and NPC1 (NB400-148; Novus Bioscience); washed, incubated with appropriate HRP-conjugated secondary antibody and developed with chemiluminescent substrate (Thermo Fisher Scientific). Images were captured on a Bio-Rad ChemiDoc XRS+ system and quantified using ImageLab software (Bio-Rad). Membranes were then stripped and re-probed with primary antibodies.

## Drug treatments and FACS analysis

Patient fibroblasts were treated with 50 $\mu$M miglustat (Actelion) 72 h, 250 $\mu$M hydroxypropyl-$\beta$-cyclodextrin (Sigma-Aldrich) 24 h and 125 $\mu$M NAD for 24 h before analysis of levels of acetylated $\alpha$-tubulin. Cells were fixed with 4% paraformaldehyde, permeabilized with FACS-perm buffer (BD Biosciences), stained with anti-acetylated $\alpha$-tubulin antibody, and analyzed on a BD FACS Canto II cytometer using FACSDiva software. 10,000 cell events were collected, and the molecules of equivalent fluorescence (MEFL) were calculated using 8-peak Rainbow calibration beads (559123; BD Biosciences).

## Electron microscopy

Cells were processed according to Eden et al (2016) (72). In brief, the cells were fixed in 2% paraformaldehyde/2% glutaraldehyde for 30 min, post-fixed in 1% osmium tetroxide and 1.5% potassium ferricyanide, and incubated in 1% uranyl acetate. The cells were then dehydrated and embedded in TAAB-812 resin. 70-nm sections were viewed on a Jeol 1010 transmission electron microscope and images captured with a Gatan Orius SC100B charge-coupled camera.

## Supplementary Information

## Acknowledgements

This research received funding from the European Union Seventh Framework Programme (FP7 2007–2013) under grant agreement no 289278 - "Sphingonet" and Action Medical Research. FM Platt is a Royal Society Wolfson Research Merit Award holder and a Wellcome Trust Investigator in Science. ME Fernández-Suárez is a Royal Society Newton International Fellow. N Platt is supported by the Wellcome Trust (202834/Z/16/Z). The Schuldiner lab is also supported by a Volkswagen foundation "Life" grant. M Schuldiner is an incumbent of the Dr. Gilbert Omenn and Martha Darling Professional Chair in Molecular Genetics.

## Author Contributions

A Colaco: formal analysis, investigation, methodology, and writing—original draft, review, and editing.
ME Fernandez-Suarez: investigation and methodology.
D Shepherd: investigation and methodology.
L Gal: investigation and methodology.
C Bibi: investigation and methodology.
S Chuartzman: investigation and methodology.
A Diot: investigation and methodology.
K Morten: methodology.
E Eden: investigation and methodology.
FD Porter: resources.
J Poulton: supervision and methodology.
N Platt: methodology and writing—review and editing.
M Schuldiner: conceptualization, resources, supervision, investigation, and writing—review and editing.
FM Platt: conceptualization, resources, supervision, funding acquisition, and writing—review and editing.

## Conflict of Interest Statement

The authors declare that they have no conflict of interest.

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
