## [Reviewer comments · Life Science Alliance]

Life Science Alliance

Unbiased Yeast Screens Identify Cellular Pathways Affected in Niemann-Pick Disease Type C

Alexandria Colaço, Maria Fernandez-Suarez, Dawn Shepherd, Lihi Gal, Chen Bibi, Silvia Chuartzman, Alan Diot, Karl Morten, Emily Eden, Forbes Porter, Joanna Poulton, Nick Platt, Maya Schuldiner, and Frances Platt

DOI: <https://doi.org/10.26508/lsa.201800253>

Corresponding author(s): Frances Platt, Oxford, University of

Review Timeline:

Submission Date:	2018-11-23
Editorial Decision:	2018-12-20
Revision Received:	2020-04-29
Editorial Decision:	2020-05-14
Revision Received:	2020-05-21
Accepted:	2020-05-22

Scientific Editor: Andrea Leibfried

Transaction Report:

December 20, 2018

Re: Life Science Alliance manuscript #LSA-2018-00253

Dr. Frances M. Platt
Oxford, University of
The Glycobiology Institute
Dept. of Biochemistry
University of Oxford
Oxford, South Parks Road OX1 3QU
United Kingdom

Dear Dr. Platt,

Thank you for submitting your manuscript entitled "Unbiased Yeast Screens Identify Cellular Pathways Affected in Niemann-Pick Disease Type C" to Life Science Alliance. The manuscript was assessed by expert reviewers, whose comments are appended to this letter.

As you will see, the reviewers appreciate the resource value of your findings, while noting that the analyses provides limited functional and mechanistic insight. The latter is not precluding publication here, and we would thus like to invite you to submit a revised version of your work. The reviewers provide constructive input on how to strengthen your work. Importantly, the robustness of the analyses needs to get improved (see specific comments of reviewer #3) and some clarifications and text alterations are needed (all reviewers). Reviewer #1 furthermore suggests some epistasis / rescue experiments that should be straightforward to perform and would add more insight. We would be happy to discuss the individual revision points further with you should this be helpful.

Thank you for this interesting contribution to Life Science Alliance. We are looking forward to receiving your revised manuscript.

Sincerely,

- A letter addressing the reviewers' comments point by point.
- An editable version of the final text (.DOC or .DOCX) is needed for copyediting (no PDFs).
- High-resolution figure, supplementary figure and video files uploaded as individual files: See our detailed guidelines for preparing your production-ready images, <http://life-science-alliance.org/authorguide>
- Summary blurb (enter in submission system): A short text summarizing in a single sentence the study (max. 200 characters including spaces). This text is used in conjunction with the titles of papers, hence should be informative and complementary to the title and running title. It should describe the context and significance of the findings for a general readership; it should be written in the present tense and refer to the work in the third person. Author names should not be mentioned.

B. MANUSCRIPT ORGANIZATION AND FORMATTING:

Full guidelines are available on our Instructions for Authors page, <http://life-science-alliance.org/authorguide>

Reviewer #1 (Comments to the Authors (Required)):

In this study, Colaco et. al. investigate the function of NPC1, a protein when mutated causes rare

lysosomal storage disorder, Neimann-Pick disease type C (NPC). To do that, they have performed three different screens using yeast as a model organism. Yeast contains homolog of NPC1 called NCR1. Authors very elegantly utilize the results obtained from screens in yeast and explore the cellular functions in NPC patient derived cells. This study is good example of using yeast as a simple model organism to address human disorders. The screens do not identify the function the NPC1 but identify many interesting pathways and cellular functions that may contribute to NPC pathogenesis. The authors attempt to link already reported mitochondrial dysfunction in NPC patients to the novel cytoskeleton phenotypes discovered from the screens. However, the manuscript in its current form lacks mechanism.

In the first screen, authors identified PMC1 and others as interacting partners of NCR1 on the vacuolar membrane. Authors validate the interaction with PMC1, which was also previously reported to interact with NPC1. However, there is no follow up to this result. This could be moved to supplement.

In the third screen, authors should validate the genetic interaction by comparing the growth of the double mutant with the single mutants from the hits. For example, authors should compare the growth of Gal1p-NCR1 fzo1delta cells with fzo1 single mutant. Minor correction required in the text where it says Gal1-NCP1 instead of Gal1p-NCR1.

Does ncr1 mutant contain mtDNA, have growth defect on respiratory media, altered distribution of mitochondria and cytoskeleton defects? Also does deletion of mitochondrial fission gene in ncr1fzo1 delta mutant rescue the growth phenotype.

Does deletion/depletion of mitofusin lead to growth defect in NPC patient cells?

It has been reported that loss of mtDNA is associated with fragmented mitochondria. Here it is reported that NPC patients cells exhibit loss of mtDNA but have elongated mitochondria. Authors should address this discrepancy.

Does supplementation with NAD, treatment of patient derived cells with miglustat or HPbetaCD reverse the mitochondrial phenotypes?

Reviewer #2 (Comments to the Authors (Required)):

NPC1 is a critical protein in the cellular trafficking of cholesterol, as well as in viral entry. Its structure was resolved in 2016. This study used yeast screening to identify novel proteins and pathways that may be associated with NPC1. Mitochondrial and cytoskeletal changes were found in NPC1 deficient cells. Overall, the study provided some useful information and table 1 may provide some value to this field. The screens in yeast appeared to be carried out very carefully. Some of the key findings were supported by mammalian experiments. One suggestion: the text mainly refers to a role for NPC1 in sphingolipid homeostasis/trafficking. However, the recent structural data have pretty much ruled out NPC1's direct role in sphingolipids. It is surprising how little these authors choose to ignore the progress made on cholesterol aspect in NPC1 function. A balanced summary and discussion of these findings in relation to cholesterol should be included, if this paper is accepted.

Reviewer #3 (Comments to the Authors (Required)):

Frances Platt and co-workers have used genetic approaches in budding yeast to identify cellular pathways affected in Niemann-Pick Disease Type C. More specifically, they have screened for interaction partners of Ncr1 (the yeast orthologue of NPC1), compensatory or redundant pathways as well as for proteins whose localization depends on Ncr1.

These screens identified several processes and pathways that may potentially contribute to NPC pathogenesis, including mitochondrial function, cytoskeleton organization, metal ion homeostasis, lipid trafficking, calcium signalling and nutrient sensing. Some of these processes (mitochondrial function and cytoskeletal function) were analyzed in NPC1 mutant cells from mouse and patients.

The data presented could provide an interesting resource for studying NPC1 function but it does not provide functional or mechanistic insight in the function of NPC1. Also the quality for some of the experiments should be improved.

Main comments:

- To strengthen the data of the synthetic lethal / sickness screen, it would be helpful to backcross the most interesting hits several times or confirm the synthetic genetic interactions in a different yeast strain, with a different genetic background and or perform complementation studies. In particular mitochondrial defects should be re-evaluated because BY4741 (and or BY4742) show increased petite frequency. This is particularly critical if the growth defects are small as in Figure 1E.
- The IP, as presented in Figure 1C, cannot be interpreted. It should include Inputs (PMCA2 and NPC1) as well as the IPed bait (NPC1). Also what is the band in rabbit IgG lane?
- The quality of the micrographs presented in Figure 1D and the description in the results sections is not very good. What are we supposed to see? I guess an increase in fluorescence signal? Nam2 appears to show only background signal. Also please show more than 2 cells.
- It is (at least for me) not possible to understand how the quantification of perinuclear mitochondria was performed. The generation of the masks appears arbitrary and there is no explanation offered on how the masks were generated. Just based on the staining of images shown in the figures it seems to me that the overall number of mitochondria with ATP-Synthase increased in the fibroblast with NPC1 mutations.
- The authors observe hyperacetylated microtubules in cells with NPC1 mutations. Given that microtubules mainly have mitotic function inside the nucleus in yeast, I find it quite a stretch to conclude that 'Moreover, our results demonstrate the conservation of cytoskeletal defects from yeast to mammals suggesting them to be hallmarks of this disease.' Please rephrase that.
- functional claims connecting the mitochondrial phenotypes with acetylated microtubules should be tested.

Minor points:

- General comment - for non-experts please explain the effects that are associated with the NPC1 patient mutations used in this study.
- Figure 1: please follow the formal yeast nomenclature.
- Figure 2C,D: It should be made clear that not total mitochondrial length was measured but length in section
- Figure 3: sizebars are missing - but the cells look very large - almost show a senescence like 'fried egg' phenotype.
- Figure 4A: please add kDa size marker.
- Figure 4C: the 'N.S.' is partially covered with the legend of the graph

Response to Reviewers Comments:

Please find below our responses to the very helpful reviewers comments in red that we have addressed in full. We are uploading a marked/highlighted copy so the changes can be easily seen, in addition to a clean version.

Reviewer1:

In this study, Colaco et. al. investigate the function of NPC1, a protein when mutated causes rare lysosomal storage disorder, Neimann-Pick disease type C (NPC). To do that, they have performed three different screens using yeast as a model organism. Yeast contains homolog of NPC1 called NCR1. Authors very elegantly utilize the results obtained from screens in yeast and explore the cellular functions in NPC patient derived cells. This study is good example of using yeast as a simple model organism to address human disorders. The screens do not identify the function the NPC1 but identify many interesting pathways and cellular functions that may contribute to NPC pathogenesis. The authors attempt to link already reported mitochondrial dysfunction in NPC patients to the novel cytoskeleton phenotypes discovered from the screens. However, the manuscript in its current form lacks mechanism.

In the first screen, authors identified PMC1 and others as interacting partners of NCR1 on the vacuolar membrane. Authors validate the interaction with PMC1, which was also previously reported to interact with NPC1. However, there is no follow up to this result. This could be moved to supplement.

Although the interaction with PMC1 was previously identified in a high throughput screen (alongside many other interactions that are false negatives) we are the first to show that this interaction is conserved to the human homologues and so feel strongly that this should remain in the main figure. In addition, we have now added some follow up data to our results by including measurements of the mRNA of mammalian ATP2B2 (PMCA2) in the NPC mouse model indicating that the loss of NPC influences the expression of the proteins it interacts with (Figure 1E).

In the third screen, authors should validate the genetic interaction by comparing the growth of the double mutant with the single mutants from the hits. For example, authors should compare the growth of Gal1p-NCR1 fzo1delta cells with fzo1 single mutant.

This is a good point; we have included the growth of the single mutants identified from the screen (Figure 2A).

Minor correction required in the text where it says Gal1-NCP1 instead of Gal1p-NCR1.

Thank you, we have corrected this to GAL1pr-NCR1.

Does ncr1 mutant contain mtDNA, have growth defect on respiratory media, altered distribution of mitochondria and cytoskeleton defects?

The ncr1 mutant does not have any gross mitochondrial defects in either morphology or growth defects on non-fermentable sources; we have included this information in the supplementary material (Supp Figure 1A,B). We were not able to fully examine cytoskeletal defects in the yeast ncr1, however this would be interesting to follow up further in a separate study.

Also does deletion of mitochondrial fission gene in ncr1fzo1 delta mutant rescue the growth phenotype.

Yes, deletion of fis1 does rescue the growth defect in the ncr1/fzo1 mutant. This has been included in Figure 2E.

Does deletion/depletion of mitofusin lead to growth defect in NPC patient cells?

We did not examine the growth rates in NPC patient cells. It is not known if there is an underlying defect in the growth in this disease. We found this to be outside of the scope of the current manuscript, although it would be interesting to examine cell division and doubling times in human and mammalian NPC disease model cells.

It has been reported that loss of mtDNA is associated with fragmented mitochondria. Here it is reported that NPC patients cells exhibit loss of mtDNA but have elongated mitochondria. Authors should address this discrepancy.

It has been described that inhibition of mitochondrial fission can lead to elongated mitochondria, resulting in loss of mitochondrial DNA, decrease in mitochondrial respiration and increase in ROS (1) – similar to what is observe in NPC. Furthermore, oxidative stress has been demonstrated to promote mitochondrial fusion, resulting in hyper-fused organelles (2). Oxidative stress has been indicated to contribute to NPC disease (3), and this may be one of the causative factors for the elongated mitochondria that we observe in the NPC patient fibroblasts.

While loss of mtDNA and fragmented mitochondria has been documented in neurodegenerative diseases, this does not exclude the possibility that reduced levels of mtDNA and hyperfused mitochondria can also be found. It is most likely that it is a disturbance to the fission/fusion dynamic balance that results in mitochondrial dysfunction.

We have now further addressed this discrepancy in our discussion by including: 'Previous studies have, however, observed fragmented mitochondria in neurodegenerative diseases (4), so taken together with our data this suggests that it is most likely a disturbance to the fission/fusion dynamic balance that results in mitochondria dysfunction.'

Does supplementation with NAD, treatment of patient derived cells with miglustat or HPbetaCD reverse the mitochondrial phenotypes?

Great questions, we are focusing on this in another study ongoing in both cells and in the NPC1 mouse model in the lab so this, therefore, is beyond the scope of the current manuscript. However, using experimental NPC therapies we have seen a reversal of some mitochondrial phenotypes.

Reviewer #2

NPC1 is a critical protein in the cellular trafficking of cholesterol, as well as in viral entry. Its structure was resolved in 2016. This study used yeast screening to identify novel proteins and pathways that may be associated with NPC1. Mitochondrial and cytoskeletal changes were found in NPC1 deficient cells. Overall, the study provided some useful information and table 1 may provide some value to this field. The screens in yeast appeared to be carried out very carefully. Some of the key findings were supported by mammalian experiments. One suggestion: the text mainly refers to a role for NPC1 in sphingolipid homeostasis/trafficking. However, the recent structural data have pretty much ruled out NPC1's direct role in sphingolipids. It is surprising how little these authors choose to ignore the progress made on cholesterol aspect in NPC1 function. A balanced summary and discussion of these findings in relation to cholesterol should be included, if this paper is accepted.

Thank you for your comments, we have included more discussion on the cholesterol aspects of NPC1 function in the introduction, specifically with reference to the recent structural data from Ncr1 and Npc2 in yeast describing how sterols/lipids in general would integrate into vacuolar membranes.

“These data implicating defects in sphingolipid and sterol trafficking are in line with the recent structural data identifying an internal hydrophobic tunnel environment in Ncr1 that would accommodate a variety of lipids, in a capture-and-shuttle mechanism (5). This yeast Ncr1 tunnel model also further supports previous work indicating that mammalian NPC1 interacts with other sterol shuttling proteins including Gram1b on the ER membrane and ORP5 on the plasma membrane, and that contact sites may be necessary for lipid export from the lysosome (6,7). Therefore, while these new models shed light on how lipids might physically move from the vacuole, the mechanisms and proteins involved in both the lipid trafficking defect and accumulation in NPC remain unknown.”

Reviewer #3 (Comments to the Authors (Required)):

Frances Platt and co-workers have used genetic approaches in budding yeast to identify cellular pathways affected in Niemann-Pick Disease Type C. More specifically, they have screened for interaction partners of Ncr1 (the yeast orthologue of NPC1), compensatory or redundant pathways as well as for proteins whose localization depends on Ncr1.

These screens identified several processes and pathways that may potentially contribute to NPC pathogenesis, including mitochondrial function, cytoskeleton organization, metal ion homeostasis, lipid trafficking, calcium signalling and nutrient sensing. Some of these processes (mitochondrial function and cytoskeletal function) were analyzed in NPC1 mutant cells from mouse and patients.

The data presented could provide an interesting resource for studying NPC1 function but it does not provide functional or mechanistic insight in the function of NPC1. Also the quality for some of the experiments should be improved.

Main comments:

To strengthen the data of the synthetic lethal / sickness screen, it would be helpful to backcross the most interesting hits several times or confirm the synthetic genetic interactions in a different yeast strain, with a different genetic background and or perform complementation studies. In particular mitochondrial defects should be re-evaluated because BY4741 (and or BY4742) show increased petite frequency. This is particularly critical if the growth defects are small as in Figure 1E.

We did not observe gross mitochondrial defects in the $\Delta ncr1$ strain (Supp Fig1), and both the control and mutant $\Delta ncr1$ strain grew normally on non-fermentable carbon sources. While the strain may show increased petite frequency we do not believe this to have negatively affected our study, particularly as mitochondrial defects were characterized further and observed in patient fibroblasts. We have also changed Figure 1E, and now show dilution assay for just Mac1 and Fzo1 with some additional controls (Figure 2A). Importantly, all strains were remade from fresh instead of back-crossing so this ensures that the phenotypes are not a result of a secondary mutation.

The IP, as presented in Figure 1C, cannot be interpreted. It should include Inputs (PMCA2 and NPC1) as well as the IPed bait (NPC1). Also what is the band in rabbit IgG lane?

This is a very good point. We have had difficulty in performing western blots on ATP2B2 due to it being a large membrane protein, and conditions that are optimal for detecting the protein in the input sample do not seem to be optimal for detecting in the pull down. We have included the input and flow through lanes (Figure 1D), as well as demonstrated why we cannot see ATP2B2 in the input lane due to the processing of the samples (boiling 90C). We are unsure of what the background band in Rabbit IgG lane is, but as we have consistently seen it in the blots we believe it to be some non-specific binding.

The quality of the micrographs presented in Figure 1D and the description in the results sections is not very good. What are we supposed to see? I guess an increase in fluorescence signal? Nam2 appears to show only background signal. Also please show more than 2 cells.

Thank you for pointing this out. We hope that we have displayed better micrographs now focusing on just Prk1 and She4 (Figure 3A). We have described the altered localization of the proteins in the figure legend.

It is (at least for me) not possible to understand how the quantification of perinuclear mitochondria was performed. The generation of the masks appears arbitrary and there is

no explanation offered on how the masks were generated. Just based on the staining of images shown in the figures it seems to me that the overall number of mitochondria with ATP-Synthase increased in the fibroblast with NPC1 mutations.

We agree that details on how the masks were created is missing. We have decided to remove this piece of data as we would like to examine this phenotype of mitochondrial distribution and how it is related to cytoskeletal function in more detail.

The authors observe hyperacetylated microtubules in cells with NPC1 mutations. Given that microtubules mainly have mitotic function inside the nucleus in yeast, I find it quite a stretch to conclude that 'Moreover, our results demonstrate the conservation of cytoskeletal defects from yeast to mammals suggesting them to be hallmarks of this disease.' Please rephrase that.

This is a very valid point, and we have rephrased that statement to read "Moreover, our results demonstrate cytoskeletal defects from actin in yeast to microtubules in humans, suggesting cytoskeletal defects may be hallmarks of this disease.

In the results section we cite that mitochondria move along actin filaments in budding yeast; yet mitochondria move along microtubules in mammalian cells.

"In higher eukaryotes, however, mitochondrial positioning is regulated by microtubules (8) rather than actin as in yeast, so we therefore examined the microtubule network in NPC deficient mammalian cells."

I hope that it is clearer that it is this connection with mitochondria movement that prompted us to move from the actin phenotype in yeast to microtubules in mammalian cells.

Functional claims connecting the mitochondrial phenotypes with acetylated microtubules should be tested.

We agree and this is an ongoing investigation, we can only hypothesize at this stage that they are connected, and we have now clarified in the discussion that this is only a hypothesis.

Minor points:

General comment - for non-experts please explain the effects that are associated with the NPC1 patient mutations used in this study. *We have explained this.*

Figure 1: please follow the formal yeast nomenclature. *Thank you – we have now changed the nomenclature as required.*

Figure 2C,D: It should be made clear that not total mitochondrial length was measured but length in section *OK*

Figure 3: sizebars are missing - but the cells look very large - almost show a senescence like 'fried egg' phenotype. *We have removed this panel from the figure.*

Figure 4A: please add kDa size marker. *Added*

Figure 4C: the 'N.S.' is partially covered with the legend of the graph. *We have now changed this so that it is clearer*

1. Parone, P. A., Da Cruz, S., Tondera, D., Mattenberger, Y., James, D. I., Maechler, P., Barja, F., and Martinou, J. C. (2008) Preventing Mitochondrial Fission Impairs Mitochondrial Function and Leads to Loss of Mitochondrial DNA. *PLoS One* **3**, e3257
2. Redpath, C. J., Khali, M. B., Drozdal, G., Radisic, M., and McBride, H. M. (2013) Mitochondrial Hyperfusion during Oxidative Stress Is Coupled to a Dysregulation in Calcium Handling within a C2C12 Cell Model. *PLoS One* **8**, e69165
3. Zampieri, S., Mellon, S. H., Butters, T. D., Nevyjel, M., Covey, D. F., Bembi, B., and Dardis, A. (2009) Oxidative stress in NPC1 deficient cells: protective effect of allopregnanolone. *J Cell Mol Med* **13**, 3786-3796
4. Knott, A. B., Perkins, G., Schwarzenbacher, R., and Bossy-Wetzel, E. (2008) Mitochondrial fragmentation in neurodegeneration. *Nat Rev Neurosci* **9**, 505-518
5. Winkler, M. B. L., Kidmose, R. T., Szomek, M., Thaysen, K., Rawson, S., Muench, S. P., Wustner, D., and Pedersen, B. P. (2019) Structural Insight into Eukaryotic Sterol Transport through Niemann-Pick Type C Proteins. *Cell*, 1-13
6. Du, X., Kumar, J., Ferguson, C., Schulz, T. A., Ong, Y. S., Hong, W., Prinz, W. A., Parton, R. G., Brown, A. J., and Yang, H. (2011) A role for oxysterol-binding protein-related protein 5 in endosomal cholesterol trafficking. *J Cell Biol* **192**, 121-135
7. Hoglinger, D., Burgoyne, T., Sanchez-Heras, E., Hartwig, P., Colaco, A., Newton, J., Futter, C. E., Spiegel, S., Platt, F. M., and Eden, E. R. (2019) NPC1 regulates ER contacts with endocytic organelles to mediate cholesterol egress. *Nat Commun* **10**, 4276
8. Fu, M. M., and Holzbaur, E. L. (2014) Integrated regulation of motor-driven organelle transport by scaffolding proteins. *Trends Cell Biol* **24**, 564-574

May 14, 2020

RE: Life Science Alliance Manuscript #LSA-2018-00253R

Dr. Frances M. Platt
Oxford, University of
The Glycobiology Institute
Dept. of Biochemistry
University of Oxford
Oxford, South Parks Road OX1 3QU
United Kingdom

Dear Dr. Platt,

Thank you for submitting your revised manuscript entitled "Unbiased Yeast Screens Identify Cellular Pathways Affected in Niemann-Pick Disease Type C". As you will see, the reviewers appreciate the introduced changes, and we would thus be happy to publish your paper in Life Science Alliance pending final revisions necessary to meet our formatting guidelines:

- Please rename the appendix tables to "supplementary Table 1" etc. and upload them in either docx or excel format
- Please list 10 authors et al in your reference list
- Please add scale bars to Figure 3B, 4A, S1A
- Please mention statistical test used and p-values in the legend for Fig 1E and 2D (also, the legend to 2A mentions one, but figure panel doesn't show any)
- Please link your ORCID iD to your profile in our submission system, you should have received an email with instructions on how to do so

A. FINAL FILES:

-- High-resolution figure, supplementary figure and video files uploaded as individual files: See our

detailed guidelines for preparing your production-ready images, <http://www.life-science-alliance.org/authors>

B. MANUSCRIPT ORGANIZATION AND FORMATTING:

Thank you for your attention to these final processing requirements.

Sincerely,

Andrea Leibfried, PhD
Executive Editor
Life Science Alliance
Meyerhofstr. 1
69117 Heidelberg, Germany
t +49 6221 8891 502
e a.leibfried@life-science-alliance.org

Reviewer #1 (Comments to the Authors (Required)):

The authors have sufficiently addressed my concerns in the revised version of the manuscript. I recommend acceptance and publication.

Reviewer #3 (Comments to the Authors (Required)):

The authors have addressed my initial concerns and improved the manuscript. It contains interesting aspects and provides an valuable resource for studying NPC1 function.

May 22, 2020

RE: Life Science Alliance Manuscript #LSA-2018-00253RR

Prof. Frances M. Platt
Oxford, University of
The Glycobiology Institute
Dept. of Biochemistry
University of Oxford
Oxford, South Parks Road OX1 3QU
United Kingdom

Dear Dr. Platt,

Thank you for submitting your Research Article entitled "Unbiased Yeast Screens Identify Cellular Pathways Affected in Niemann-Pick Disease Type C". It is a pleasure to let you know that your manuscript is now accepted for publication in Life Science Alliance. Congratulations on this interesting work.

DISTRIBUTION OF MATERIALS:

Again, congratulations on a very nice paper. I hope you found the review process to be constructive and are pleased with how the manuscript was handled editorially. We look forward to future exciting

submissions from your lab.

Sincerely,
